# OpenReview forum: "LLMail-Inject: A Dataset from a Realistic Adaptive Prompt Injection Challenge"
_NeurIPS.cc/2025/Datasets_and_Benchmarks_Track — Submitted to NeurIPS 2025 Datasets and Benchmarks Track_

### Official Review · Reviewer_naTw · 2025-06-28

**Rating:** 5
**Confidence:** 4

**Summary:**

This paper presents LLMail-Inject, a public challenge simulating realistic adaptive prompt injection attacks, where participants inject malicious instructions into emails to trigger unauthorized tool calls in an LLM-based email assistant. The challenge spanned multiple defense strategies, LLM architectures, and retrieval configurations across two phases, resulting in a dataset with unique attack submissions. The authors provide a comprehensive analysis of defenses (such as Spotlighting, Prompt Shield, LLM-Judge, and TaskTracker) and attack strategies, release the challenge code, full dataset, and analysis, which offer new insights into the instruction-data separation problem. This work serves as a foundation for future research on practical and structural solutions to prompt injection, addressing gaps in understanding defense comparisons against adaptive adversaries and the complexity of attacking real-world retrieval systems.

**Additional Feedback:**

NA

**Dataset Code Accessibility:**

Yes

**Dataset Code Comments:**

The dataset is in https://huggingface.co/datasets/microsoft/llmail-inject-challenge.

**Ethical Comments:**

No significant ethical concerns remain.

**Ethical Considerations:**

No, there are no or only very minor ethics concerns

**Final Justification:**

Thanks for dealing with my concerns. I hope the authors will include the additional information in the paper to make it stronger.

**Limitations Weaknesses:**

This dataset should be valuable, but I have some minor concerns:

1. Model-Specific Limitations: It insufficiently demonstrates the dynamic adaptability of attackers in response to evolving defenses, and its reliance on a limited range of models (with a focus on Phi-3) restricts the generalizability of its findings across diverse LLM architectures. In some defense work [1,2], different models have significant differences in defense

[1] Xu, Zhao, Fan Liu, and Hao Liu. "Bag of tricks: Benchmarking of jailbreak attacks on llms." arXiv preprint arXiv:2406.09324 (2024).

[2] Liu, Zichuan, et al. "Protecting your llms with information bottleneck." NeurIPS (2024): 29723-29753.

2. Dataset Annotation Biases: 179,458 submissions (of 208,095) lack ground-truth labels and rely on an "LLM-annotator". The prompt for annotation (Appendix J) admits that social engineering attacks may be misclassified as "Unclear."  Labeling inaccuracies may affect benchmark reliability, especially for indirect/social engineering attacks. Suggest manually annotating and comparing a small portion of the sampled data (and releasing a verified subset with high-confidence labels).

**Strengths Contributions:**

Strengths:
1. Novel realistic challenge design: LLMail-Inject stands out as a public challenge simulating a realistic LLM-integrated email assistant scenario, where participants craft adaptive prompt injections to trigger unauthorized tool calls. This design bridges the gap between theoretical defenses and real-world applications, as it incorporates retrieval systems and multiple defense strategies—elements critical to understanding end-to-end attack complexity.
2. Large-scale dataset: The dataset has 208,095 unique attack submissions, including annotations (29K labeled injections), metadata, and synthetic benign emails for benchmarking. This dataset captures diverse attack strategies and adaptive tactics against known defenses, offering unprecedented insights into how attackers bypass protections.
3. Systematic evaluation of defenses: Tested state-of-the-art defenses (e.g., Spotlighting, Prompt Shield, LLM-Judge, TaskTracker), revealing their effectiveness.
4. Highlighted trade-offs: Preventive defenses (e.g., Spotlighting) reduced utility in long-context settings.

---

> ### Author Rebuttal · Authors · 2025-07-29
>
> Thank you so much for your positive review and encouraging words. We are very glad by your acknowledgment. **Bridging the gap between theoretical defenses and real-world applications is exactly one of the motivation of organizing this challenge**.
>
> We reply to your comments below.
>
> > Model-Specific Limitations: It insufficiently demonstrates the dynamic adaptability of attackers in response to evolving defenses, and its reliance on a limited range of models (with a focus on Phi-3) restricts the generalizability of its findings across diverse LLM architectures. In some defense work [1,2], different models have significant differences in defense
>
> At the time of the competition, The Phi-3 model we used was the latest one released. We focused on the Phi family as we used TaskTracker probe that was trained on Phi. However, we also tested on GPT-4o, which according to the analysis in [1], performs the best among closed models on most benchmarks.
>
> Unfortunately, studying adaptive attacks against defenses/models faces this dilemma. In order to capture the state of defenses/models, one has to tailor the attacks against them. Ensuring full generalization to other/future defenses may sacrifice customizing attacks against the studied ones. However, we believe the dataset can be valuable for other models. After the paper deadline and the release of Meta SecAlign model, we ran it on our dataset. While other benchmarks achieve a very low attack success rate (3-4%), ours achieves 32%.
>
> We are in the process of considering how to maintain this platform and update it with new levels. **We also released our platform and competition code in order to facilitate running further challenges by the community**. During the competition, we received many requests from participants who enjoyed participation, and from professors who thought the challenge is a good introduction to LLM security. Running similar challenges with closed-models with APIs (without GPUs) and to a smaller-scale audience can be affordable and feasible to adapt.
>
> > Dataset Annotation Biases: 179,458 submissions (of 208,095) lack ground-truth labels and rely on an "LLM-annotator". The prompt for annotation (Appendix J) admits that social engineering attacks may be misclassified as "Unclear." Labeling inaccuracies may affect benchmark reliability, especially for indirect/social engineering attacks. Suggest manually annotating and comparing a small portion of the sampled data (and releasing a verified subset with high-confidence labels).
>
> Thank you for your comments. We are currently considering how to improve this limitation. We will update the dataset accordingly. Potential automated annotation methods include:
>
> - Use the defenses as multiple judges. Examples annotated as attacks by one or more defense can be indicated as "more likely to be attacks"
> - Use the attacks as user prompts directly while prompting the model to execute any instructions in them if any; if such prompts result in calling the tool, then they can be labelled as valid attacks.
> - We will more thoroughly evaluate the annotations on a random subset and report agreement.
>
>
> [1] Chen et al. "Meta SecAlign: A Secure Foundation LLM Against Prompt Injection Attacks." arXiv preprint arXiv:2507.02735 (2025).

---

### Official Review · Reviewer_NFxA · 2025-07-01

**Rating:** 3
**Confidence:** 4

**Summary:**

This paper present the results of LLMail-Inject, a public challenge that gathered adaptive indirect prompt 249 injection attacks against various defenses. The benchmark is designed to align with the realistic scenario, but too many existing attack and defense methods are not discussed or evaluated.

**Dataset Code Accessibility:**

Yes

**Dataset Code Comments:**

The codes and detailed readme file are provided

**Ethical Considerations:**

No, there are no or only very minor ethics concerns

**Final Justification:**

The rebuttal provided by authors only partly addressed my concerns, the evaluation and disscusion of existing defense methods reqires a throughly revision. Furthermore, the benchmark solely focus on email assistant use case, the versatility is still relatively limited.

**Limitations Weaknesses:**

1. The benchmark proposed in this paper primarily designs a scenario in which an email AI assistant is hijacked through prompt injection. Admittedly, this is a highly realistic scenario. However, I believe that for a benchmark paper, this focus appears overly narrow, as prompt injection has a wide range of applications and can theoretically target any AI assistant designed for specific scenarios, not just those limited to email. Therefore, I am concerned about whether this benchmark can comprehensively evaluate defense methods.

2. The prompt injection task has been investigated for a while, diverse SOTA attack and defense methods have  been proposed. This manuscript lacks of discussion and evaluation on these methods.

3. Although this manuscript has introduced five defense strategies, a detailed and comprehensive review of both exiting attack and defense methods is missing. The below references are only a fraction of existing methods, there should be more papers had been published, the authors are encouraged to review these related papers.


[1] Agarwal, Divyansh, et al. “Prompt Leakage Effect and Mitigation Strategies for Multi-Turn LLM Applications.” Proceedings of the 2024 Conference on Empirical Methods in Natural Language Processing: Industry Track, edited by Franck Dernoncourt et al., Association for Computational Linguistics, 2024.

[2] Guo, Jiawei, and Haipeng Cai. System Prompt Poisoning: Persistent Attacks on Large Language Models beyond User Injection. 1, arXiv:2505.06493, arXiv, 10 May 2025.

[3] Liu, Yi, et al. Prompt Injection Attack against LLM-Integrated Applications. arXiv:2306.05499, arXiv, 2 Mar. 2024.

[4] Liu, Yupei, et al. Formalizing and Benchmarking Prompt Injection Attacks and Defenses. 2024.

[5] Peng, Yu, et al. “RepeatLeakage: Leak Prompts from Repeating as Large Language Model Is a Good Repeater.” Proceedings of the AAAI Conference on Artificial Intelligence, vol. 39, no. 25, 25, Apr. 2025.

[6] Perez, Fábio, and Ian Ribeiro. Ignore Previous Prompt: Attack Techniques for Language Models. 2022.

[7] Wen, Tongyu, et al. Defending against Indirect Prompt Injection by Instruction Detection. 1, arXiv:2505.06311, arXiv, 8 May 2025.

[8] Yi, Jingwei, et al. “Benchmarking and Defending against Indirect Prompt Injection Attacks on Large Language Models.” Proceedings of the 31st ACM SIGKDD Conference on Knowledge Discovery and Data Mining V.1, 2025.

[9] Zhang, Yiming, et al. Effective Prompt Extraction from Language Models. 2024.

**Strengths Contributions:**

1. This paper introduce a prompt injection challenge that aligns well with the realistic scenario.

---

> ### Author Rebuttal · Authors · 2025-07-29
>
> Thank you for the time you spent reviewing our paper.
>
> > The benchmark proposed in this paper primarily designs a scenario in which an email AI assistant is hijacked through prompt injection. Admittedly, this is a highly realistic scenario. However, I believe that for a benchmark paper, this focus appears overly narrow, as prompt injection has a wide range of applications and can theoretically target any AI assistant designed for specific scenarios, not just those limited to email. Therefore, I am concerned about whether this benchmark can comprehensively evaluate defense methods.
>
> - Thank you for observing this is a highly realistic scenario.
>
> - While it would indeed be ideal to evaluate "a wide range of applications", it would be impractical from an engineering perspective (and taking into consideration the capacity of participants as well) to organize a challenge with the similar depth as LLMail-Inject yet expanding to many cases.
>
> - As mentioned before, **we provided 64 unique challenges**, which kept the participants occupied for months.
>
> - Finally, **we kindly request the reviewer to consider the specific wording of the NeurIPS Datasets and Benchmarks Tracks CfP**, which asks for "In-depth analyses of machine learning challenges and competitions (by organisers and/or participants) that yield important new insight". **LLMail-Inject was an in-depth study of a highly realistic attack scenario against LLM applications, as the reviewer acknowledged**.
>
> - Please check our reply to reviewer jBuu and xFxN for more details. In comparison to previous benchmarks, **our dataset is significantly larger (200K unique attacks vs. ~1K in previous benchmarks), more diverse (no fixed attack templates are used), and more challenging even for SoTA defenses released after our paper (other benchmarks show <4% on Meta SecAlign, ours shows 32% without any additional tweaking)**. The email setup is a strong feature of our work, rather than a weakness, as it is more challenging for defenses and leads to contextual and complex attacks.
>
> > The prompt injection task has been investigated for a while, diverse SOTA attack and defense methods have been proposed. This manuscript lacks of discussion and evaluation on these methods.
>
> - The defenses we included (LLM Judge, Prompt Shield, Task Tracker, Spotlight, GPT-4o trained with instruction hierarchy) captured a large variety of LLM defenses and importantly covered broad categories (text-based classifiers, LLM as a detector, internal states detector, prompting defenses, and training defenses); unfortunately, for this specific challenge, we had to limit their number: in total, **this resulted in 64 individual scenarios, and it would have been impractical (and strenuous on the participants) to include more.**
>
> - However, it is our hope to use the same platform to include more defenses in future challenges. We also release the platform to enable the community to adapt our code base for new defenses.
>
> - We are unclear why the reviewer is asking for a comparison of existing attacks: in this challenge, **attacks were provided by the participants**. Further, *evaluating existing attacks in this setup would break the adaptive claim*.
>
> - Thank you for providing a large number of references, which we will include and discuss in the camera-ready version of this paper. Please note that some of these references are not related to prompt injection (for example, some on System Prompt Poisoning). **Additionally, some defenses appeared on Arxiv a week before NeurIPS deadlines (Defending against Indirect Prompt Injection by Instruction Detection). The preparation for this competition started in September 2024**.

---

> > ### Author Response · Authors · 2025-08-05
> > **Kindly let us know if you have questions**
> >
> > Dear Reviewer NFxA,
> >
> > We appreciate your time and feedback on our work. As the deadline of the discussion period is approaching, we would like to kindly check whether our responses above have sufficiently addressed your concerns and assisted you in re-evaluating our work.
> >
> > If you have any further comments or questions, please do not hesitate to let us know before the discussion deadline.
> >
> > Best regards

---

> ### Comment · Reviewer_NFxA · 2025-08-06
>
> Thank you for the detailed answer, my concerns have been partly addressed. I have raised my score to 3 (weak reject). However, a more comprehensive evaluation require a throughly revision. Furthermore, the benchmark solely focus on email assistant use case, the versatility is still relatively limited (reviewer jBuu also acknowledge this). Thus I am unable to further increase my score.

---

> > ### Author Response · Authors · 2025-08-08
> > **Could you please give us more information?**
> >
> > Dear reviewer NFxA,
> >
> > Thanks a lot for your comment. May we please ask you to kindly let us know what you envision as a more comprehensive evaluation so that we can include it in a revision? The competition invites adaptive attacks; therefore, if the design of the competition includes attacks, it would have invalidated our claim of adaptive attacks. Additionally, we are unable to clearly identify specific other defenses that our paper and competition are lacking. For example, many of the references you provided are related to prompt extraction attacks, which are different than indirect prompt injection. Please also note that we use many representatives of defenses, including production-level ones such as Prompt Shields.
> >
> > We would also appreciate it if you could please comment on our rebuttal in our reply to reviewer jBuu and xFxN in which we discuss the email assistant use case. In summary: 1) defenses and models we tested are agnostic to email assistants, showing broad applicability of our dataset, 2) emails are a challenging testbed for both defenses and models, 3) this is a highly practical scenario of real-world data exfiltration attacks, 4) SoTA models which show saturating performance on other benchmarks still significantly more frequently execute the attacks in our dataset.

---

### Official Review · Reviewer_xFxN · 2025-07-03

**Rating:** 3
**Confidence:** 3

**Summary:**

This paper introduces a new dataset, LLMail-Inject, designed to study indirect prompt injection attacks in realistic email scenarios. The proposed framework involves five key components: the attacker, the user, the email database, the large language model (LLM), and the defense mechanisms. The setup also explores four progressive difficulty levels, varying along three dimensions: the retrieval configuration, the positioning of the attacker's email within the mailbox, and the total number of emails present.

**Dataset Code Accessibility:**

Yes

**Ethical Considerations:**

No, there are no or only very minor ethics concerns

**Final Justification:**

The authors have partially addressed my concerns regarding the novelty. After reading other reviewers’ comments and considering the narrow scenarios of the benchmark, I will keep my score.

**Limitations Weaknesses:**

1. The novelty of LLMail-Inject is not sufficiently justified. Lines 27–29 mention that the community lacks an established understanding of two factors, but this alone does not clearly establish what differentiates LLMail-Inject from existing datasets or benchmarks. It would strengthen the paper to explicitly compare LLMail-Inject with existing resources and highlight its unique contributions in terms of design, scale, realism, or task complexity.

2. Section 3 would benefit from the inclusion of a statistical summary, such as a table or histogram, that provides key dataset metrics (e.g., number of emails, distribution of attack types, number of users, etc.). This would help readers quickly grasp the scope and structure of the dataset.

3. Figure 2, which currently presents statistical information, could be improved by replacing it with pie charts. These alternatives are more visually effective and commonly used for presenting categorical statistics clearly and professionally.

4. The related work section (Section 6) is too brief and lacks sufficient coverage of existing literature. To provide a comprehensive context for LLMail-Inject, the authors should expand this section to include and discuss more relevant works on prompt injection attacks, particularly those in indirect or email-based scenarios.

5. Figure 1 does not effectively illustrate the overall pipeline of indirect prompt attacks within the email environment. As a central figure in the paper, it should provide a clear and comprehensive overview of the attack flow, including the interactions between the attacker, user, email client, retrieval process, and LLM.

**Strengths Contributions:**

1. The authors consider a realistic and practical email-based environment for studying indirect prompt injection attacks, which enhances the relevance of the proposed benchmark to real-world applications.

2. The proposed dataset incorporates adaptive attack configurations, which demonstrate a thoughtful design that reflects evolving adversarial strategies.

3. The paper conducts a comprehensive evaluation of various large language models and defense mechanisms using the proposed attack dataset. This extensive experimentation allows the authors to explore the behavior of LLMs under indirect prompt injection attacks in diverse settings. Based on these results, the paper presents several insightful observations and takeaways that could inform future research in both offensive and defensive strategies for LLM security.

---

> ### Author Rebuttal · Authors · 2025-07-29
>
> Thank you for the time you spent reviewing our papers and your positive comments regarding the realistic and practical email-based environment, the adaptive attack configurations, and our observations and takeaways. We address your comments below, and would be happy to provide further details should you require further clarifications to fairly assess our work.
>
> > The novelty of LLMail-Inject is not sufficiently justified. Lines 27–29 mention that the community lacks an established understanding of two factors, but this alone does not clearly establish what differentiates LLMail-Inject from existing datasets or benchmarks. It would strengthen the paper to explicitly compare LLMail-Inject with existing resources and highlight its unique contributions in terms of design, scale, realism, or task complexity.
>
> Thank you for this comment. The updated version of our paper will include a discussion as per the reviewer's recommendation. We summarize it here:
>
> - AgentDojo [1] is one of the now well-established benchmarks for indirect prompt injection attacks in a simulation environment of tool use. AgentDojo has **only 629** security test cases (e.g., making an unauthorized transaction), augmented with **fixed general-purpose templates** for attacks (e.g., inserting this is an important message). Compared to it, we specifically invite **attacks that are tailored against each defense** instead of generic attacks.
>
> - SEP [2] is another benchmark to evaluate instruction-data separation. It uses only **100** probes as injected instructions and vary the combination of users' tasks and data paragraphs into which these probes are injected. INJECAGENT [3] is another benchmark for tool use that has only **1,054** test cases. BIPIA dataset [4] has overall **250** attack prompts. Compared to all previous work, **we provide a significantly larger dataset of over 200K attacks**.
>
> - **Task Complexity**: Our task involves RAG; an attacker has to get their email to be retrieved by being more relevant to the query compared to other benign emails, adding more complexity.
>
> - **Diversity**: Instead of using fixed attack templates (which participants found to be more easily detected), the collected dataset has more diverse attacks. As we simulate an email assistant client, the setup is inherently challenging for existing defenses (please refer to reply to reviewer jBuu), inviting future defenses that can discern between the nuanced contexts of attacks vs. benign emails.
>
> - **Challenging defenses**: We also develop new defenses in the competition (the LLM-judge) which was found to be the hardest one to break with 0.96 to 0.99 recall rate and takes on average 506.2 trails to break (please see Table 2).
>
> - **Performance on current benchmarks is saturating**: *After NeurIPS deadline*, Meta SecAlign [5] was released which is a model that is trained to resist prompt injection attacks. The paper [5] shows very low attack success rate (<4\%) on all benchmarks. **On LLMailInject, the ASR is 32\%** (preliminary results without much tuning and exploration regarding the location of the attack within the context window).
>
> [1] Debenedetti et al. "Agentdojo: A dynamic environment to evaluate prompt injection attacks and defenses for llm agents." Advances in Neural Information Processing Systems 37 (2024): 82895-82920.
>
> [2] Zverev et al. "Can LLMs Separate Instructions From Data? And What Do We Even Mean By That?." The Thirteenth International Conference on Learning Representations.
>
> [3] Zhan et al. "InjecAgent: Benchmarking Indirect Prompt Injections in Tool-Integrated Large Language Model Agents." Findings of the Association for Computational Linguistics ACL 2024. 2024.
>
> [4] Yi et al. "Benchmarking and defending against indirect prompt injection attacks on large language models." Proceedings of the 31st ACM SIGKDD Conference on Knowledge Discovery and Data Mining V. 1. 2025.
>
> [5] Chen et al. "Meta SecAlign: A Secure Foundation LLM Against Prompt Injection Attacks." arXiv preprint arXiv:2507.02735 (2025).
>
> > Section 3 would benefit from the inclusion of a statistical summary—such as a table or histogram—that provides key dataset metrics (e.g., number of emails, distribution of attack types, number of users, etc.). This would help readers quickly grasp the scope and structure of the dataset.
>
> > Figure 2, which currently presents statistical information, could be improved by replacing it with pie charts. These alternatives are more visually effective and commonly used for presenting categorical statistics clearly and professionally.
>
> > The related work section (Section 6) is too brief and lacks sufficient coverage of existing literature. To provide a comprehensive context for LLMail-Inject, the authors should expand this section to include and discuss more relevant works on prompt injection attacks, particularly those in indirect or email-based scenarios.
>
> > Figure 1 does not effectively illustrate the overall pipeline of indirect prompt attacks within the email environment. As a central figure in the paper, it should provide a clear and comprehensive overview of the attack flow, including the interactions between the attacker, user, email client, retrieval process, and LLM.
>
> Thank you for these suggestions. We are happy to include that in the final version which will have an additional page to expand the related work and provide a statistical summary.

---

> > ### Author Response · Authors · 2025-08-04
> > **Kindly let us know**
> >
> > Dear Reviewer xFxN,
> >
> > We appreciate your time and feedback on our work. We also thank you for the automatic acknowledgment. As you have read our rebuttal, we would like to kindly check whether our responses above (**including the new experiment we ran on a SoTA model** to demonstrate the advantage of our benchmark) have sufficiently addressed your concerns and assisted you in re-evaluating our work.
> >
> > If you have any further comments or questions, please do not hesitate to let us know before the discussion deadline.
> >
> > Best regards

---

> > ### Comment · Reviewer_xFxN · 2025-08-06
> >
> > I appreciate the authors' response. I will maintain my current score at this stage and take some time to further discuss with the other reviewers in the next phase.

---

### Official Review · Reviewer_jBuu · 2025-07-03

**Rating:** 3
**Confidence:** 2

**Summary:**

The submission introduces LLMail-Inject, a multi-level adversarial prompt injection challenge and associated dataset for email assistant agents. It simulates realistic end-to-end attacks—retrieval, defense bypass, tool invocation, and parameter extraction—over two phases with over 460K raw submissions. The paper details the challenge design, defenses evaluated (Spotlighting, Prompt Shield, LLM Judge, TaskTracker), dataset statistics, quantitative analysis of defense effectiveness, and common attacker strategies. The primary contribution is the release of a large, richly annotated benchmark for prompt-injection research.

**Dataset Code Accessibility:**

Yes

**Dataset Code Comments:**

The dataset is available at https://huggingface.co/datasets/microsoft/llmail-inject-challenge

**Ethical Considerations:**

No, there are no or only very minor ethics concerns

**Final Justification:**

I appreciate the scope constraints for a competition paper and the clarification on challenge design and practical relevance. I have also considered other reviewers’ opinions. However, the core concerns that drove my evaluation score remain unaddressed. Given this, I’m keeping my score unchanged.

**Limitations Weaknesses:**

The focus is solely on an email assistant use case. Broader LLM applications (chatbots, code assistants, document QA) are not evaluated, limiting generality. Although a large dataset, its utility beyond email-centric research is unclear, and the paper does not demonstrate transferability to other domains.

Computational overheads (especially for LLM Judge) are noted qualitatively but lack quantitative runtime benchmarks in realistic deployments.

Real-world agents often involve human review; the benchmark assumes fully automated execution, which reduces ecological validity.

**Strengths Contributions:**

Models retrieval, detection, and tool invocation in a single pipeline, exposing multi-stage attack vectors

Clear figures and tables with informative captions; the paper is concise and easy to follow.

461 K+ raw attempts, 208 K+ deduplicated prompts, multiple difficulty levels, two LLMs (Phi-3, GPT-4o) across phases, with rich metadata for each record. Also, an Empirical evaluation of each defense’s recall, attacker success rates per level, and cost–accuracy trade-offs (Figures 2–4).

Dataset and code are openly available

---

> ### Author Rebuttal · Authors · 2025-07-29
>
> Thank you so much for your positive feedback and acknowledging the strength of our setup, including the multi-stage attack vector, the dataset size, the rich metadata, and the empirical evaluation. We address your comments below, and happy to provide more details should you require further clarifications to fairly assess our work.
>
> > The focus is solely on an email assistant use case. Broader LLM applications (chatbots, code assistants, document QA) are not evaluated, limiting generality. Although a large dataset, its utility beyond email-centric research is unclear, and the paper does not demonstrate transferability to other domains.
>
> We highlight the following:
>
> - **D&B scope calls for competitions**: We urge the reviewer to consider the NeurIPS Datasets and Benchmarks Tracks CfP, which specifically asks for **"In-depth analyses of machine learning challenges and competitions (by organisers and/or participants) that yield important new insight"**.
>
> - **Feasible/fair competitions**: It would be unfair to ask for a comprehensive security evaluation of all LLM applications in a single challenge; this would have been impractical from an engineering perspective -- deploying this challenge required the efforts of several engineers for multiple months, and it would have not allowed to obtain the same depth of insights that our analysis achieved.
>
> - **Competition scale**: We also remark that the challenge involved many scenarios, and totalled a set of 64 unique levels.
>
>  - **Practical relevancy**: We note such email assistant use case itself is extremely practical and relevant for real-world applications [1].
>
>  - **Dataset size and insights**: Thank you for remarking on the size of the dataset that was produced by this competition; this is the first dataset of its size, and it is the only existing dataset that enabled comparing SOTA defenses against *adaptive* attackers. Specifically, **it enabled measuring for the first time how hard it is (in terms of the number of queries) to break a system** that resembles true real-world scenarios.
>
>  - **Agnostic to defenses**: We point out that while all scenarios in this challenge involved (different setups of) prompt injection attacks via email, **defenses were not specific to these attacks**. This further gives credit to the utility of this setup beyond email settings.
>
>  - **Emails are challenging**: **Focusing on email assistants is rather a strong feature of our work that challenges existing defenses and models**. Benign emails may contain conversations, requests, or questions (a phrase such as "please confirm your attendance by replying to this email"), which is directed to the recipient instead of the assistant; both defenses and models must treat these as such. As we discussed, many classifiers had a very high False Positive rate on these benign emails. Additionally, unlike clearly out-of-context prompt injections, attacks in our dataset can be contextual and highly resemble benign emails (please refer to Figure 1). Current defenses and models overfit on obvious out-of-context attacks.
>
> > Computational overheads (especially for LLM Judge) are noted qualitatively but lack quantitative runtime benchmarks in realistic deployments.
>
> We release the output of the LLM Judge, from which, the computational overhead (i.e., number of tokens) can be easily calculated.
> We will include this in an updated version of this paper.
> We also disclose all the details of the models we used (e.g., GPT-4o for the judge). In practice, such computational overheads may vary depending on the model's size and, if reasoning models are used, the allowed reasoning tokens budget.
>
> > Real-world agents often involve human review; the benchmark assumes fully automated execution, which reduces ecological validity.
>
> There are, alas, many cases where humans are not involved in deciding whether to make a tool call or not. There are many real-world applications with increasing autonomy (e.g., Cursor). Human-in-the-loop systems
> may hinder performance and it could be desirable to minimize the number of required human reviews.
>
> On the other hand, looking at tool calls in LLMail-Inject was a proxy to reliably measure successful prompt injection attacks;
> without them, one would need to automatically parse the LLM's natural language output to verify if the injection succeeded or not, and this is error-prone.
>
> [1] Embrace The Red, wunderwuzzi's blog, "Microsoft Copilot: From Prompt Injection to Exfiltration of Personal Information"

---

> > ### Author Response · Authors · 2025-08-05
> > **Kindly let us know if you have questions**
> >
> > Dear Reviewer jBuu,
> >
> > We appreciate your time and feedback on our work. As the deadline of the discussion period is approaching, we would like to kindly check whether our responses above have sufficiently addressed your concerns and assisted you in re-evaluating our work.
> >
> > If you have any further comments or questions, please do not hesitate to let us know before the discussion deadline.
> >
> > Best regards

---

> > > ### Comment · Area_Chair_p1ke · 2025-08-07
> > > **Please respond to author rebuttal**
> > >
> > > Reviewer jBuu: Would you please take a minute to clarify if the author rebuttal has addressed your concerns or not? This will help the authors respond and help me to better understand your concerns in the context of the rebuttals and other reviews. For example, can you suggest another dataset that does this task (or a similar one) better? The use of automated evaluation is a limitation, but wouldn't human evaluation have limitations as well (such as higher cost and less reproducibility)? What would it take for this dataset to be worthy of publication? Thanks!

---

> > ### Comment · Reviewer_jBuu · 2025-08-09
> >
> > Thanks for the detailed rebuttal and for releasing the dataset. I appreciate the scope constraints for a competition paper and the clarification on challenge design and practical relevance. I have also considered other reviewers’ opinions. However, the core concerns that drove my evaluation score remain unaddressed. Given this, I’m keeping my score unchanged.

---

### Note · Authors · 2025-08-12

Dear reviewers, ACs, and SACs,

We thank the reviewers again for their time and feedback.

To facilitate final discussions, we summarize below our main contributions and the rebuttal's outcome.

## **Our competition**

We presented LLMail-Inject, a public challenge simulating realistic adaptive prompt injection attacks, where participants inject malicious instructions into emails to trigger unauthorized tool calls in an LLM-based email assistant.

The challenge spanned multiple defenses, LLM architectures, and retrieval configurations across two phases, spanning a total of 64 levels. We systematically used and compared state-of-the-art defenses including production-level ones. Participants had to adapt and tailor their strategies to different defenses and LLMs.

The attack's goal is complex; an attacker has to get their email retrieved, evade defenses, trigger the tool call with the right format and arguments, and leak data from the model's context window. Our work enabled measuring for the first time how hard it is (in terms of queries) to fully break a system that resembles true real-world scenarios; with that we aim to inform developers.

The email assistant usecase we used is highly relevant in practice. It is nuanced and challenging for many detection defenses, as they must differentiate between benign conversations directed to the user and instructions directed to the assistant, otherwise, utility would be highly impaired.

## **Our dataset**
The challenge resulted in a dataset with over 200K unique and diverse attacks, which is the first of its kind. Other benchmarks include ~1000 attacks or less using mostly fixed attack templates.

While other benchmarks show 0-4% success rate on state-of-the-art models (released after NeurIPS deadline), our dataset shows 32%.

We open-source the dataset, our analysis code, and the competition platform.

## **Planned changes**
Based on the reviewers' comments, we will thoroughly discuss the novelty over previous work and the unique setup of the challenge. We are also committed to improving the dataset's annotation as kindly suggested by Reviewer naTw.

## **Final remarks**
Given the importance of this problem for LLM security and the complexity of our dataset, we believe our work will serve as a highly useful benchmark for future defenses. We also believe in the importance of public red teaming to understand the limitations of defenses and inform both the public and developers of critical vulnerabilities.

---

### Decision · Program_Chairs · 2025-09-18

**Decision:**

Reject

**Comment:**

This paper describes a competition run in conjunction with SaTML 2025, in which teams competed to trick email-processing LLM agents into doing certain tasks. Several hundred teams competed. The dataset includes detailed information on how the competition was run, including the defenses that the attackers had to overcome, and the results from all teams. The associated paper has some analysis on the competition, its results, winning strategies, and lessons learned.

The reviewers for this paper criticized it for being narrowly focused in that it only looked at email, but email is not a narrow application and is very important and a major fraud/attack vector, so the AC and SAC reject their criticism. Reviewers also suggested that these attacks would fail in contexts where human approval is required before an agent takes action, but the AC and SAC override that criticism as there are extensive automated email agent systems in use, and completely automated agents are becoming more commonplace. Reviewers were asked to expand on their criticisms, but the response was limited.

We believe this paper aligns well with what the dataset and benchmarks track is meant to do, which is to encourage, acknowledge, and disseminate excellent work that improves our understanding and measurement of important ML problems. (In fact, the SaTML competition from 2024 was also published in the NeurIPS datasets and benchmarks track.)

We believe the lessons learned in this competition should be shared more widely, and that there may be more that can be learned from the dataset itself. The results here represent the work of several hundred teams.

(P.S. The references list has some capitalization problems, where "llm" and "satml" aren't properly capitalized. This is trivial to fix.)

===== FINAL UPDATE FROM DB Track PCs ====

The final decision for this paper has been taken by the program chairs after consultation with the SACs. All Senior Area Chairs have ranked papers according to the feedback from the AC during the review process. We decided to leave the original meta-review to reflect the opinion of the AC in light of the initial discussions with reviewers and SAC.